# Environmental Indicator Plants in Mountain Forests: A Review

**DOI:** 10.3390/plants13233358

**Published:** 2024-11-29

**Authors:** Lucian Dinca, Vlad Crisan, Gruita Ienasoiu, Gabriel Murariu, Romana Drasovean

**Affiliations:** 1National Institute for Research and Development in Forestry “Marin Dracea”, Eroilor 128, 077190 Voluntari, Romania; dinka.lucian@gmail.com (L.D.); novac_baba2001@yahoo.com (G.I.); 2Forestry Faculty, University Stefan Cel Mare of Suceava, 13 Strada Universitații, 720229 Suceava, Romania; 3Department of Chemistry, Physics and Environment, Faculty of Sciences and Environmental, Dunarea de Jos University Galati, Domneasca Street No. 47, 800008 Galati, Romania; gmurariu@ugal.ro (G.M.); romana.drasovean@ugal.ro (R.D.)

**Keywords:** journals, articles, keywords, VOSviewer, indicator plants, bibliometric review

## Abstract

Plant indicators are important in studies related to the environment, and mountain forests are rich in such plants. We conducted this study using the Web of Science Core Collection tools and the VOSviewer program combined with a classic review, based on the specialty literature. The topic of indicator plants in mountain forests has been (the first article published in a prestigious journal dates back to 1980) and remains relevant (after 2015, between 40 and 60 articles were published annually), with the total number of publications found being 665 articles and 22 review articles. These publications were authored by researchers from 96 countries, the most representative being the USA, China, and Germany (which also have large areas of mountain forests), and were published in 306 journals, with the most important being Ecological Indicators, Forest Ecology and Management, Forests, Journal of Vegetation Science, and Plant Ecology. They belong to the main scientific fields of Ecology, Forestry, Plant Sciences, and Environmental Sciences. The most frequently used keywords are vegetation, diversity, biodiversity, and forests. Their evolution over the past decade shows that the focus has shifted from keywords specific to this topic to those related to biodiversity and conservation, and more recently to climate change. Indicator plants in mountain forests are extremely varied but can be used successfully in the monitoring activity. Climatic conditions or human interventions lead to the dynamic of these indicator plants.

## 1. Introduction

Plant indicators, also known as bio-, phyto-, or environmental indicators, are species that typically exist within a specific, narrow environmental range [1]. The ideal indicator species can reliably signify a distinct set of environmental conditions [2,3,4].

Additionally, a species acting as an indicator should consistently thrive under a unique set of environmental factors and not be found outside those conditions. An ideal indicator frequently signals distinct environmental states [3] and can reflect changes in the biological status of a given ecosystem. These species are valuable tools for assessing ecosystem health. Consequently, indicator plants are precious natural resources that, if managed wisely, can be preserved for future generations [5].

Plants are gaining more and more importance in the objective evaluation of a healthy environment. For example, they can be a very good indicator of air pollution—a fact that is mentioned by many authors [6,7,8]. In addition, indicator plants have been identified for pH and water levels in peatlands [9]. 

Plant species predict the presence of specific mineral reserves. Because of this, indicator plants are used in geobotanical methods of mineral exploration (with a role in indicating the presence of aluminum boron, cobalt, copper, gold, iron, lead, manganese, nickel, selenium, silver, uranium or zinc) [10,11].

However, indicator plants can also be used in studying some animal species, such as the ecology of land mollusks [12], or in monitoring the growth of pest populations [13].

Indicator plants are extensively employed for monitoring various environmental factors [14,15], and they offer information about one site faster than physical or chemical measurements. Also, they provide a good and quick measure of the site conditions [16]. 

Species are selected as indicators when they (i) represent the biotic or abiotic status of their environment, (ii) provide insights into the effects of environmental changes, or (iii) predict the presence of other species, taxa, or communities in a specific region [17,18,19]. These species are often identified by analyzing their occurrence or abundance across different sites and grouping these locations based on shared characteristics [20,21].

Mountains are characterized by rapid shifts in environmental factors and vegetation over short distances, which has led to extensive research on the link between plant distribution and environmental conditions [22,23,24]. 

Bibliometric analysis is a contemporary approach for evaluating research through essential bibliometric principles. It employs statistical methods to assess, describe, and visualize the literature in relevant fields [25]. This technique can outline the current state of knowledge, offer new insights into characteristics, and anticipate trends in specific areas of study [26,27,28,29]. In essence, it enables researchers and policymakers to efficiently grasp the fundamentals of a field and identify emerging trends. The core elements of bibliometric analysis include (1) qualitative and quantitative assessments of publications indexed in databases utilizing statistical and computational tools, (2) collaboration among different journals, countries, institutions, authors, and co-occurrence categories, and (3) keyword tracking and dynamic analysis. This analytical approach has been extensively applied in literature reviews across information science, environmental science, agriculture, microbiology, geography, and various other disciplines [30,31,32,33].

Bibliometrics involves the quantitative evaluation of data to identify publication trends and research patterns within a particular field of study. This analysis examines data to explore collaboration networks among authors, scientific outputs and their impacts, contributions from institutions, and research contributions from different countries, among other factors [34].

A large number of bibliometric review articles have been published. A search on WOS shows 6890 articles since 1985, as of 30 August 2024, compared to 6450 articles on 14 June 2024, indicating an impressive increase in just 45 days. The most significant growth occurred between 2020 and 2023, with 1000 to 3000 articles published annually.

Regarding plant species, numerous articles have been published to date on endangered plants [35], invasive plants [36], plant response to high temperature and drought [37], and on certain species, such as *Cannabis sativa* L. [38], Moringa oleifera Lam. [39,40], maize [41], strawberry [42], *Punica grantum* L. [43], *Cucumis melo* L. [44], and *Cucumis sativus* L. [45].

Despite the significance of both indicator plants and mountain regions, no existing review or bibliometric study has focused on publications concerning this subject in the specialized literature. To address this gap, our objective was to qualitatively analyze the patterns of scientific publications (including research articles and review papers) related to this topic over the past fifty years. This article aims to perform a review (bibliometric + classic) of publications concerning indicator plants in mountain forests, covering the distribution of publication types, principal scientific fields, publication years, authorship, countries of origin, journals featuring the articles, the primary keywords employed, the species that can be included in the indicator plants category from mountain regions, and their dynamic and usage in monitoring actions.

## 2. Results

### 2.1. A Bibliometric Review

Regarding this topic, up until 2023, 716 publications had been identified. The distribution of these is as follows: 665 articles (93% of total publications), 24 proceeding papers (3%), 22 review articles (3%), and 5 book chapters (1%).

In terms of the scientific fields of the published articles, the most representative are Ecology (203 articles), Forestry (163 articles), Plant Sciences (162 articles), and Environmental Sciences (149 articles).

The first published paper on this topic was published in a prestigious scientific journal in 1980. Since then, the number of publications has steadily increased, with a significant surge especially after 2015 (Figure 1). Citations have followed a similar upward trend, mirroring the increase in published articles. Citation data: citing articles = 15,236, times cited = 16,610, average per item = 23.73 [46].

A total of 201 authors have been identified, each of whom has published at least two articles on this topic. The authors with the most articles are Jeorg Ewald, Jean-Luc Dupouey, Sheikh Marifatul Haq, and Jean-Claude Gegout, each with seven articles.

Researchers from 96 countries on five continents have contributed to articles on this topic (Figure 2).

The most represented countries are the USA (132 articles), China (116 articles), and Germany (109 articles). Following them are a group of European countries with 38–46 articles: France, Poland, Czech Republic, and Italy. Countries can be grouped into five clusters: the first one and also the largest includes Italy, Poland, Spain, Norway, Netherlands, Canada, Czech Republic, and Austria; the second includes Germany, England, Switzerland, France, and Iran; the third includes USA, China, Russia, and Japan; the fourth includes Belgium, Turkey, Georgia, India, and Pakistan; and the fifth, the smallest, includes Hungary and Croatia (Figure 3).

The most representative institutions for authors who have published on this topic are as follows: Chinese Academy of Science (with 53 articles), Russian Academy of Science (with 25 articles), University of Chinese Academy of Science (with 21 articles), and United States Department of Agriculture (with 20 articles).

The majority of articles (98%) are written in English, but there are also articles in 10 other languages, such as German (eight articles), Russian (seven articles), Croatian (five articles), and Spanish (five articles).

Articles on this topic have been published in 306 journals, with the most contributions coming from *Ecological Indicators* (37 articles), *Forest Ecology and Management* (30 articles), and *Forests* (22 articles). However, if total link strength is considered, the top three are: *Journal of Vegetation Science*, *Forest Ecology and Management*, and *Plant Ecology* (Table 1).

Journals can be grouped into three clusters (with at least five journals per cluster): cluster 1 includes *Ecological Engineering*, *Ecological Indicators*, *Frontiers in Ecology and Environment*, *Journal of Plant Ecology*, *Paleogeography Paleoclimatology Paleoecology*, and *Science of the Total Environment*; cluster 2 includes *Biodiversity and Conservation*, *Mountain Research and Development*, *Pakistan Journal of Botany*, *Restoration Ecology*, *Scientific Reports*, and *Scientific World Journal of Science*; cluster 3 includes *Ecogeography*, *Ecological Applications*, *Ecosphere*, *Journal of Environmental Biology*, and *Journal of Ethnobiology* (Figure 4).

The most representative publishers are Elsevier (167 articles), Wiley (104 articles), and Springer Nature (101 articles).

The most frequently used keywords are vegetation, diversity, biodiversity, and forests (Figure 5 and Table 2). Keywords can be grouped into four clusters: the first includes diversity, forests, richness, species richness, landscape, classification, and Ellenberg indicator values; the second includes vegetation, climate, dynamics, ecosystems, disturbance, land-use, regeneration, climate change, and pollen; the third includes patterns, forest, mountain, growth, ecology, plants, nitrogen, and carbon; and the fourth includes biodiversity, indicators, management, and plant diversity.

In 2015, the predominant keywords were specific to this topic (vegetation, forests, mountains), while in 2016–2017, keywords such as biodiversity, diversity, conservation, restoration, and management became more common. In 2018–2019, the focus shifted to climate, climate change, and growth (Figure 6).

### 2.2. A Classical Review

#### 2.2.1. Plant Species Used as Environmental Indicators in Mountain Areas

By analyzing the specialty literature, we have identified the following species of indicator plants from mountain forests (Table 3).

#### 2.2.2. Environmental Indicator Plants in Mountain Forests and Monitoring

Indicator plants are an important part of monitoring activities. It is believed that communities of mountain plants are sensitive to climate changes, which allows them to indicate these effects sooner [61]. Analyzing biochemical and biophysical vegetation variables is essential in developing climate change studies and biodiversity monitoring indicators [62]. The biodiversity loss that appears in mountain ecosystems requires integrated approaches in order to enhance monitoring under the pressure of human activities [63].

The ground-floor vegetation belongs to the ‘Intensive Forest Monitoring Programme’ of ICP Forests. Also known as ‘Level II monitoring’, it was launched to gather detailed information about ecosystem processes while covering a larger spatial scale compared with classical ecosystem processes [64]. Plant populations are good indicators for soil acidity and nutrients—two essential aspects correlated with current changes in site conditions [65,66]. In addition, ground-floor vegetation has an essential role in determining plant biodiversity, due to the fact that a large part of total plant diversity is connected to the forest floor [67,68]. Similar to other ecosystem layers, the herb layer’s floral composition depends on numerous natural and anthropogenic factors. In order to understand floral composition and its evolution, studies must approach a large number of parameters from a wide range of ecological domains. These include soil chemistry, atmospheric deposition or meteorological conditions [66]. In this way, such an approach is clearly different from other studies that focus on comparing floristic compositions across decades [69,70,71], which lack exact locations and measurements and should consider essential environmental factors.

Starting from the hypothesis that given the natural range of temporal variability of vegetation, a significant portion of that variability is attributed to the effects of climatic variation, some authors have studied the time variation of vegetation from mountain areas (e.g., Ababneh and Woolfenden in White Mountains of California, USA [72]), establishing adequate monitoring methods. Similarly, other authors (Bassler et al. [73]) have studied which vascular plant species from the Bavarian Forest National Park would respond to climate changes. They have suggested a three-level monitoring program in order to effectively track the vegetation’s response to climate changes.

Focusing on regeneration and monitoring, Bauer and Bergmeier [74] have analyzed plant communities from Crete’s mountain woodlands. They have found relevant associations by using a set of exclosures at different heights and locations.

#### 2.2.3. Dynamics of Environmental Indicator Plants in Mountain Forests

In most cases, the dynamics are low. For example, Lenoir et al. [75] have analyzed plant community changes caused by climate warming in the Jura Mountains in 1989–2007. They reached the conclusion that changes at a community level are dependent on climate warming and the specific dynamics of stands. Compared to the local increased temperatures, the species distribution did not vary so much. A similar insignificant dynamic was also observed in France in the case of European beech mixed forests, Norway spruce and silver fir. These were replaced by stands dominated by spruce species in the montane area of the northern Calcareous Alps. In other parts of the Alps, this should have resulted in a replacement of native vegetation by coniferous species, with an increase in plants indicating acidity and nitrogen levels. However, this fact is not valid for France [76]. In a similar way, Merle et al. [77] studied the change in vegetation from a Mediterranean massif (Penyagolosa, Spain) over 46 years. They saw that altitudinal shifts in species distribution were present in only 14% of plant species. However, in other parts, the dynamic was constant or even growing. The Dhofar Mountains in Oman are such an example, where forest cover is stable and slightly increasing (Arnold) [78].

In the Šumava Mountains, Czech Republic, changes in plant species composition have shown a decrease since the 1970s. This applies to the cover and/or frequency of hygrophilous and sciophilous species (e.g., *Athyrium distentifolium*, *Oxalis acetosella*) in climax spruce forests. This indicates the fact that waterlogged spruce forests prefer a stand that is drier or has better light (e.g., *Calamagrostis villosa*, *Vaccinium vitis-idaea* or *Avenella flexuosa*) (Wild) [79].

Csortek et al. [80] have also analyzed plant species composition changes over the last 92 years, but in the Western Tatra Mountains. They have observed that the main factors causing these changes are represented by livestock grazing and windthrows. On the other hand, in Spain, low- and moderate-intensity donkey grazing improved plant diversity while decreasing mountain pasture biomass (Seggara et al.) [81]. Another factor that leads to changes in vegetation composition is represented by changes in atmospheric nitrogen. This was analyzed by McDonnell et al. [82] in the Shenandoah and Great Smoky Mountain National Parks, USA.

For the mountain ranges of Central Spain, an analysis of future potential tree-species distributions and shifts in plant communities under climate change indicates that the most significant transformations will occur at piedmont and lower mountain elevations, driven by increased water deficits [83].

## 3. Discussion

Of course, most publications on this topic are articles, but we must also note the significant number of reviews published over time. Zolotova et al. [84], analyzing modern research based on Ellenberg indicator values, concluded that the majority of studies examined, among other things, the influence of various factors on plants. Since part of the chosen topic refers to mountain areas, some review articles address this component [85,86]. Other articles focus on methods for plant inventory, such as monitoring [66] and remote sensing [87]; the structure of plant communities in certain geographical regions [88]; or on the evolution of vegetation in certain geological eras [89,90]. Also well-represented are review articles that analyze the influence of climate change on plants [91,92,93].

In terms of the scientific fields to which the published articles belong, there is a predominance of Ecology over other fields such as Forestry or Plant Science, as reflected in the article titles. Naturally, Ecology encompasses vast areas, including topics from other scientific fields, but the number of articles classified within Ecology has become increasingly substantial [94,95,96].

The trend observed in other bibliometric analyses—namely the exponential increase in published articles in recent years [97,98,99]—is also true for indicator plants in mountain forests. There are two explanations for this phenomenon: on the one hand, the significant increase in the number of scientific journals and published articles in recent times, and on the other hand, the growing interest of researchers in the analyzed subject.

Since this topic refers to mountain forest areas, it is natural that the countries of the published authors are those with significant forested mountain areas. For instance, 304,022,000 hectares of the United States are forested; 33% of China’s surface is mountainous; Germany has 11,419,124 hectares of forested land; France has 17.1 million hectares; the Czech Republic has 2,648,000 hectares; and in Italy, 76.8% of the territory is covered by mountains [100].

Regarding the journals where articles on this topic have been published, many have terms such as vegetation or plants in their titles (e.g., *Journal of Vegetation Science*, *Applied Vegetation Science*, *Vegetation History and Archaeobotany*, *Plant Biosystems*, *Flora*), or they include ecology terms, which aligns with the primary scientific fields discussed above (e.g., *Forest Ecology and Management*, *Plant Ecology*, *Ecological Indicators*).

The keywords chosen for the article titles are among the top keywords used by authors who have published on this topic: indicator ranks ninth, mountains ranks sixth, and forest ranks fifth. However, diversity ranks first, and biodiversity ranks third, which underscores the crucial role that indicator plants play in determining biodiversity [49,101,102]. Other top-ranking keywords include patterns (fourth), dynamics (seventh), species richness (eighth), and communities (ninth). Many authors have analyzed plant indicator patterns [103,104] and their dynamics [74,105,106].

Keywords related to climate also hold a special place: climate ranks 12th, climate change ranks 13th, and climate change (a typographical variant) ranks 17th. If these last two terms were combined, they would likely rank sixth. Many authors have drawn connections between indicator plants and climate change [73,83,107].

An analysis of keyword usage over time shows that in 2016–2017, terms such as biodiversity, diversity, conservation, restoration, and management became more prevalent. This shift reflects the growing concern of researchers with biodiversity and conservation efforts. For the period 2018–2019, keyword analysis indicates an increased focus on climate change.

Indicator plant species from mountain forests are very diverse: from 14 articles that analyze these species on four continents, 98 species were inventoried (Table 3), with not even two species repeating. These indicator plant species differ based on the geographical area, climate or field orography [108,109,110], leading to this high diversity.

Of course, monitoring actions from mountain areas must include an inventory of indicator plants. But, unlike other biotope components, they record the speed of the changes caused by environmental factors [111,112,113]. Due to this fact, researchers use them the most. However, the high volume of data produced by inventorying and identifying them causes an impediment in using them at a larger scale.

The dynamics of indicator plants from mountain forests are variable—they are low in most cases but are sometimes high (especially in arid ecosystems [78] or vulnerable ones [114,115]). The causes that lead to this dynamic are mainly climatic (global warming, strong winds that cause wind throws etc.) or human (livestock grazing).

## 4. Materials and Methods

A comprehensive bibliometric analysis necessitates an extensive compilation of databases related to studies on the effects of high temperatures and drought stress on plants. We utilized the Web of Science Core Collections from the Web of Science (WOS) database [59] to build a bibliographic resource on plant responses to these conditions. WOS is widely recognized by researchers and was frequently employed as a valuable tool in recent bibliometric analyses [116,117,118]. Specifically, using the “Advanced Search” feature in WOS, we queried fields and information based on Boolean operators, with the terms “high temperature stress” AND “drought stress”, conducting a literature search spanning from 1 January 2008 to 31 December 2021. The search was performed for “All Fields” within the Web of Science Core Collection. Additionally, we enhanced the precision of keywords via the “Exact Search” option.

A bibliometric study was carried out to evaluate the global scientific output on indicator plants in mountain forests from 1980 to 2023. This assessment used the Science Citation Index-Expanded—Web of Science database, retrieving 716 publications. The online SCI-Expanded database was searched with the terms “indicator plants in mountain forests” and “mountain forests and their indicator plants” to compile relevant research papers from 1996 to the present. The study focused on ten key areas: (1) types of publications, (2) Web of Science categories, (3) publication years, (4) countries, (5) institutions, (6) language, (7) journals, (8) publishers, (9) authors, and (10) keywords.

Data were processed using the Web of Science Core Collection tools [119], Excel [120], Geochart [121], and VOSviewer version 1.6.20 [46], which provide features like visualization maps and cluster analysis. Only research articles and review papers were considered in the present study. A classic review was also realized by analyzing numerous articles published about this subject. The data were grouped into three large result categories: plant species used as indicators in mountain areas; indicator plants in mountain forests and monitoring; and dynamics of indicator plants in mountain forests.

## 5. Conclusions

The 665 articles and 22 review articles published between 1980 and 2023 on indicator plants in mountain forests span numerous scientific fields, with the most important being Ecology, Forestry, Plant Sciences, and Environmental Sciences. These articles were authored by numerous researchers, with 201 authors having published at least two articles. The authors come from 96 countries, with the most represented being countries with large areas of forest, including the USA, China, and Germany. The vast majority of articles (98%) are written in English and have been published in 306 journals, with the most important appearing in *Ecological Indicators*, *Forest Ecology and Management*, and *Forests*. Based on total link strength, key journals also include *Journal of Vegetation Science*, *Forest Ecology and Management*, and *Plant Ecology*. The principal publishers are Elsevier, Wiley, and Springer Nature. The most frequently used keywords are vegetation, diversity, biodiversity, and forests. The journals are grouped into three clusters, and the keywords are grouped into four clusters. An analysis of the evolution of keyword usage over time reveals a shift (since 2015) from keywords specific to this topic (vegetation, forests, mountains) to terms related to biodiversity and conservation (2016–2017), such as biodiversity, diversity, conservation, restoration, and management, and later to terms related to climate change (2018–2019). The species that represent indicator plants in mountain forests are extremely numerous, with variables based on the geographic area or climate. Indicator plants are an important aspect of monitoring activities as they record changes caused by environmental factors faithfully and rapidly. The dynamics of indicator plants from mountain forests, caused by climatic or human factors, are mostly low and sometimes high. What we can learn from this review is that environmental indicator plants have an important role in mountain forest ecosystems, but not only. This aspect was studied by many authors, with many plant species identified as indicators of different environmental aspects. These species were also used in the monitoring activity. However, there are still many aspects that should be analyzed in regard to indicator plants, such as the following: identifying other plant species that can indicate certain ecological aspects from mountain forest areas; finding new links between indicator plants and some environment aspects; and correlations between indicator plants and the vegetation’s historical evolution from the mountain area.

## Figures and Tables

**Figure 1 plants-13-03358-f001:**
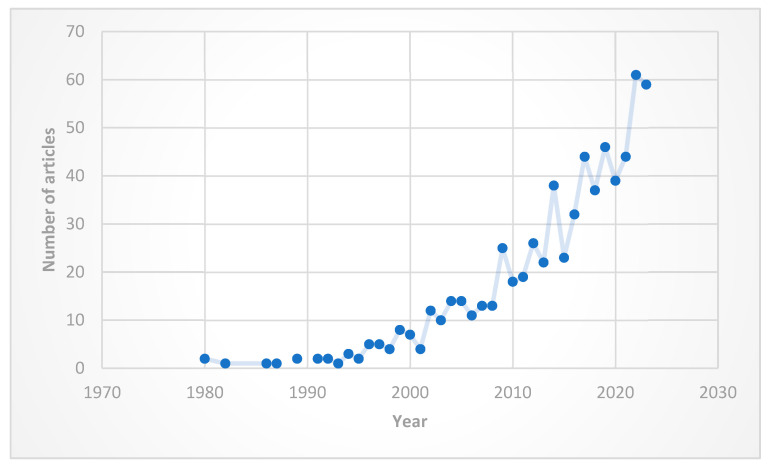
Distribution of publishing papers per year.

**Figure 2 plants-13-03358-f002:**
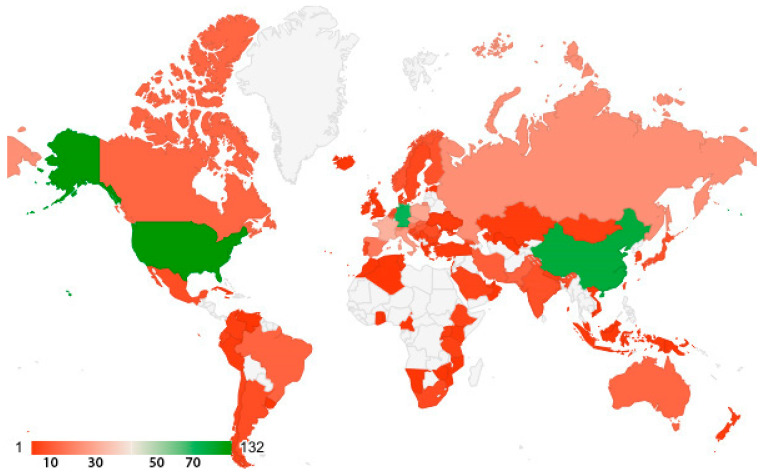
Countries with authors contributing to articles on environmental indicator plants in mountain forests.

**Figure 3 plants-13-03358-f003:**
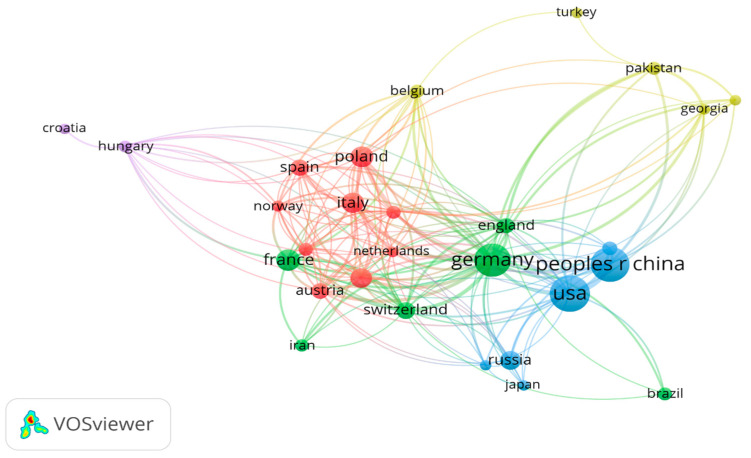
Clusters of countries with authors of articles on environmental indicator plants in mountain forests. The node size and thickness of the connecting lines are proportional to the number of documents assigned to each country. The connections represent the collaboration network among research institutions.

**Figure 4 plants-13-03358-f004:**
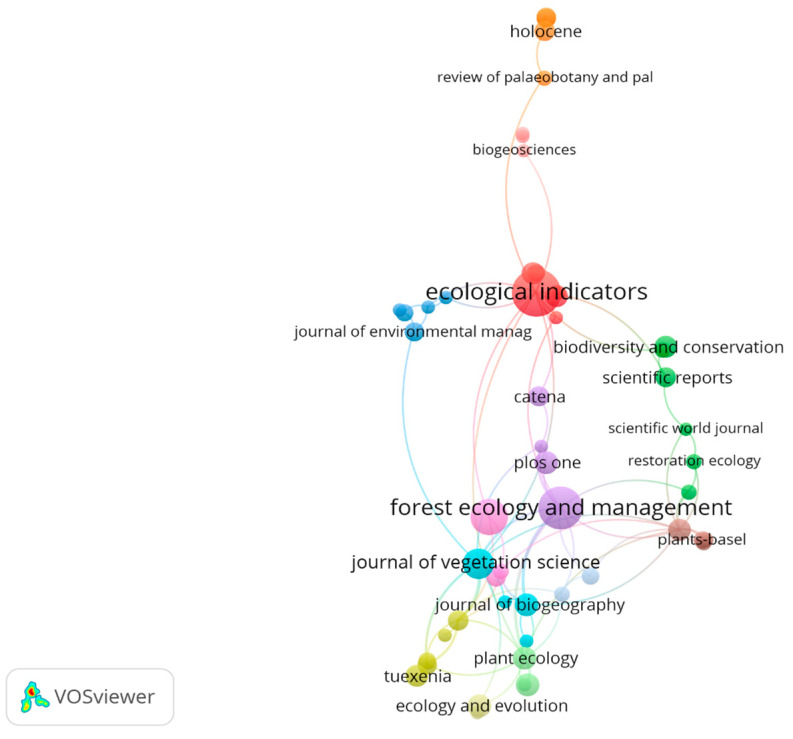
The main journals where articles on indicator plants in mountain forests have been published.

**Figure 5 plants-13-03358-f005:**
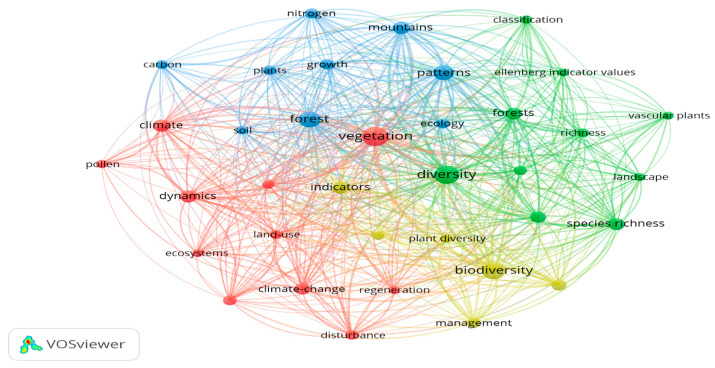
Authors’ keywords concerning the environmental indicator plants in mountain forests. The node size and thickness of the connecting lines are proportional to the number of documents in which the keyword appears. The colors indicate the cluster the item belongs to, and the connection line between nodes represents co-occurrence; the shorter the distance between the different nodes, the stronger the relationship between the keywords.

**Figure 6 plants-13-03358-f006:**
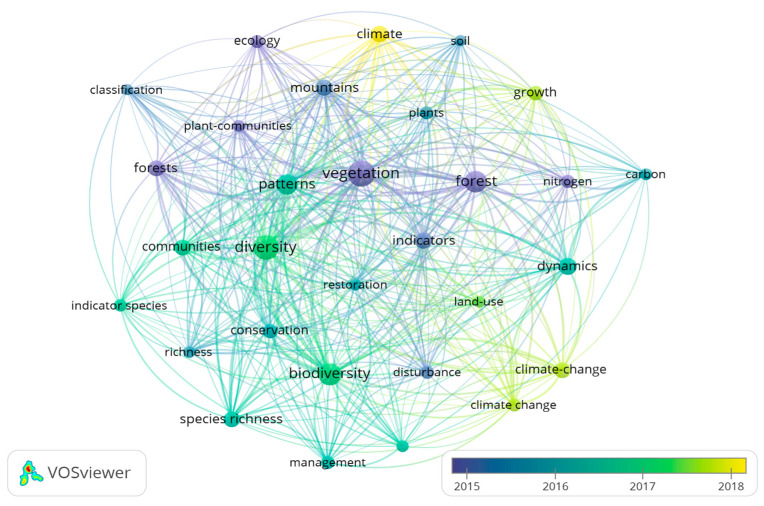
Annual distribution of keywords on indicator plants in mountain forests. The node size and thickness of the connecting lines are proportional to the number of documents containing the keyword. The colors indicate the cluster to which the item belongs, and the connection line between nodes represents co-occurrence; the shorter the distance between nodes, the stronger the relationship between the keywords.

**Table 1 plants-13-03358-t001:** The most representative journals publishing articles on environmental indicator plants in mountain forests.

Crt. No.	Review	Documents	Citations	Total Link Strength
1	*Journal of Vegetation Science*	15	301	12
2	*Forest Ecology and Management*	30	990	11
3	*Plant Ecology*	9	330	11
4	*Applied Vegetation Science*	7	97	9
5	*Ecological Indicators*	37	672	8
6	*Folia Geobotanica*	6	272	7
7	*European Journal of Forest Research*	8	97	6
8	*Journal of Biogeography*	9	455	5
9	*Holocene*	7	324	3
10	*Plant Biosystems*	6	97	3
11	*Biologia*	5	87	2
12	*Ecology and Evolution*	9	120	2
13	*Flora*	5	76	2
14	*Forests*	22	85	2
15	*Journal of Environmental Management*	6	165	2
16	*Phytoeconologia*	9	71	2
17	*Vegetation History and Archaeobotany*	6	133	2

**Table 2 plants-13-03358-t002:** The most frequently used keywords in articles on environmental indicator plants in mountain forests.

Cr. No.	Keyword	Occurrences	Total Link Strength
1	diversity	122	369
2	vegetation	136	343
3	biodiversity	101	303
4	patterns	82	227
5	forest	94	211
6	mountains	57	172
7	dynamics	58	161
8	species richness	52	157
9	communities	49	147
10	indicators	57	147
11	forests	52	141
12	conservation	44	134
13	climate	57	128
14	climate-change	49	120
15	restoration	29	110
16	richness	29	107
17	management	36	101
18	climate change	36	97
19	disturbance	34	96

**Table 3 plants-13-03358-t003:** Species of environmental indicator plants identified in different mountain areas from around the globe.

Crt. No.	Geographic Area	Indicator Plants	Source
1	Karst mountains of Pingguo County, Guangxi, China	*Apluda mutica*, *Microstegium nodosum*, *Miscanthus floridulus*, *Chromolaena odoratum*, *Lepionurus sylvestris*, *Asplenium sampsoni*, *Asplenium saxicola*	Ou et al., 2020 [47]
2	Qilian Mountains, northwest China	*Potentilla parvifolia. Kobresia myosuroides*, *Potentilla saundersiana*, *Saussurea japonica*, *Agropyron cristatum*, *Anemone cathayensis*, *Ranunculus tanguticus*, *Elymus nutans*.	Huo et al., 2014 [48]
3	Western Himalaya	For lower elevation (2450–2800 m): *Pinus wallichiana*, *Sambucus weightiana*, *Impatiens bicolour.* For middle elevation (2900–3300 m): *Abies pindrow*, *Betula utilis*, *Achillea millefolium.* Higher elevations (above 3400 m): *Rheum austral*, *Sibbaldia cuneata*, Bergenia *strachyei*, *Aster falconeri*, *Iris hookeriana*, *Ranunculus hirtellus.*	Khan et al., 2014 [49]
4	Kashmir Himalayan in India	Dry habitat: *Geranium wallichianum*, Senecio *chrysanthemoides*, *Rumex dentatus*, *Artemisia absinthium*, *Delphinium cashmerianum*, *Filipendula vestit*, *Lavatera kashmiriana*, *Persicaria amplexicaulis*, *Viburnum grandiflorum. * Wet habitat*: Impatiens glandulifera*, *Veronica persica*, *Ranunculus hirtellus.*	Haq et al., 2022 [50]
5	Hindu Kush Mountains	*Celtis caucasica-Lonicera hispida-Pennisetum orientale*, *Juglans regia-Rubus idaeus-Tragopogon gracilis*, *Abies pindrow-Parrotiopsis jacquemontiana-Agrostis gigantea*.	Anwar et al., 2023 [51]
6	Hyrcanian region of northern Iran	*Trisetum bungei*, *Polygonum hyrcanicum*, *Ranunculus amblyolobus*	Naqinezhad et al., 2019 [52]
7	Caucasus, Georgia	*Brachypodium sylvaticum*, *Festuca drymeja*	Goginashvili et al., 2021 [53]
8	Acipayam district, Turkey	*Berberis crataegiana*, *Lonicera etrusca var. etrusca*, *Juniperus feoettidissima*, *Phlomis armeniaca.*	Gülsoy and Özkan, 2013 [54]
9	Mountain Parnonas, Greece	*Geranium molle*, *Cerastium candidissimum*, *Vicia villosa*, *Euphorbia myrsinites*, *Odontarrhena muralis*, *Medicago lupulina*, *Lotus corniculatus*, *Crepis fraasii*, *Bellis sylvestris*, *Trifolium stellatum*	Solomou et al., 2023 [55]
10	Solling mountains, NW Germany	*Atrichum undulatum* in beech forests and the mosses *Campylopus flexuosus* and *Dicranum polysetum* in spruce forests	Schmidt et al., 2006 [56]
11	Bavarian Alps	indicator plant species of nutrient-poor forest sites: *Potentilla erecta*, *Vaccinium vitis-idaea*, *Homogyne alpina*, *Huperzia selago*	Reger et al., 2014 [57]
12	Taita Mountains, Kenya	*Tabernaemontana stapfiana*, *Macaranga conglomerata*, *Strombosia scheffleri*, *Craibia zimmermannii*, *Newtonia buchananii*, *Albizia gummifera*, *Rapanea melanophloeos*, *Syzygium sclerophyllum*	Aerts et al., 2011 [58]
13	Gurage Mountains, Ethiopia	*Olinia rochetiana*, *Maytenus addat*, *Ilex mitis*, *Maesa lanceolata*, *Vernonia rueppellii*	Seta et al., 2019 [59]
14	Bosque Protector Chongón Colonche, Ecuador	*Erythrochiton giganteus*, *Phytelephas aequatorialis*, *Matisia grandifolia*, *Clavija eggersiana*, *Compsoneura mutisii*, *Caryodaphnopsis theobromifolia*, *Chrysochlamys dependens*, *Prestoea decurrens*, *Couepia platycalyx*	Jadán et al., 2022 [60]

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
