# Peer review of "Environmental Indicator Plants in Mountain Forests: A Review"

_plants, 2024, doi:10.3390/plants13233358_

Round 1

Reviewer 1 Report

Comments and Suggestions for Authors

This manuscript is an excellent review of indicator plants in temperate forests. The review is very well done with a large number of cited articles and a good analysis on the subject. It only has typographical errors, and for example in Fig. 1 the legend is misformatted. In Fig. 2, line 105 it says ...main ten scientific, it is better to write ten than 10. In Fig. 3, line 114 it says ...of articles, it would probably be better to say ...of publishing papers. In Fig. 4, it is better to divide the scale that goes from 1 to 132, writing intermediate numbers.

Finally, and probably not the authors' concern, in lines 51-53 there is a frequently made error: Although there are thousands of books and published research that claim that altitude influences many ecological functions, this is not true. It is very easy to say altitude, but in reality what changes with altitude is climate and possibly the soil as well. Therefore, saying that altitude is an ecological factor is an artifact. The responses may be temperature, precipitation, etc.

Author Response

Coments 1-in Fig. 1 the legend is misformatted.

Response 1-We have eliminated this graph at the request of another reviewer.

Coments 2-In Fig. 2, line 105 it says ...main ten scientific, it is better to write ten than 10.

Response 2-We have eliminated this graph at the request of another reviewer.

Coments 3-In Fig. 3, line 114 it says ...of articles, it would probably be better to say ...of publishing papers.

Response 3-Thank you for pointing this out. We have made the change.

Coments 4-In Fig. 4, it is better to divide the scale that goes from 1 to 132, writing intermediate numbers.

Response 4-Agree. We have, accordingly, added intermediary values on the graph’s scale.

Coments 5-in lines 51-53 there is a frequently made error: Although there are thousands of books and published research that claim that altitude influences many ecological functions, this is not true. It is very easy to say altitude, but in reality what changes with altitude is climate and possibly the soil as well. Therefore, saying that altitude is an ecological factor is an artifact. The responses may be temperature, precipitation, etc.

Response 5-Indeed, the altitude is not the main factor, but temperatures, precipitation, etc. In this regard we have eliminated this paragraph, including the citations from the References section.

Reviewer 2 Report

Comments and Suggestions for Authors

The topic chosen, indicator plants in mountain forests, is not clearly of general interest. The term "indicator plants" is ambiguous, (that is, indication of what?) and mountain forests itself is unclear, as many mountain areas are characterized by much nonforested areas.

The methodology is fine, in terms of being a standard search for literature. But the documentation is exhaustive without being very helpful. In particular, I question if Figures 1, 2, and 5 are useful, or if Table 1 should be included. There is lots of detail presented about the various authors, but since the general importance of the topic is unclear, those details do not seem helpful.

It is striking in Figure 4 how many important mountainous areas in the world are not included. Because of this, the details on authorship and journals published in seem to be unhelpful.

Given that the methods used are fine, a successful revision would rethink the terms and concepts utilized to see if indicator plants need to be defined more specifically (indicators of what?) and if the need for forest cover is important when looking at this topic in terms of global mountainous regions.

Author Response

Coments 1-The term "indicator plants" is ambiguous, (that is, indication of what?) and mountain forests itself is unclear, as many mountain areas are characterized by much nonforested areas.

Response 1-Thank you for pointing this out. We agree with the comments. Therefore, we have changed the title and text with the term “environmental indicator plants”. Indeed, we now think that the term is more clear. However, we do not want to change the term “mountain forests”. This is the area that we have studied, the area that has specific plants that are different from other mountain areas, such as alpine meadows. As such, we did not refer to the entire mountain area in our study, but only to the one covered by forests.

Coments 2-I question if Figures 1, 2, and 5 are useful, or if Table 1 should be included.

Response 2-We have eliminated Figures 1 and 2, but we consider that Figure 5 and Table 1 are useful (as you can see country clusters, while the connections between them are harder to render in writing. In addition, the first 17 journals where articles were published are more visible, while the text mentions just the first 3).

Coments 3-It is striking in Figure 4 how many important mountainous areas in the world are not included. Because of this, the details on authorship and journals published in seem to be unhelpful.

Response 3-In this figure, only uncoloured countries (countries from Europe without mountains like Estonia, Letonia, Lithuania and Belarus; a part of Africa - especially the Saharan area; Asia or Greenland) do not have authors that have published on this topic. We improved the graph with stronger colors that are more visible.

Coments 4-Given that the methods used are fine, a successful revision would rethink the terms and concepts utilized to see if indicator plants need to be defined more specifically (indicators of what?) and if the need for forest cover is important when looking at this topic in terms of global mountainous regions.

Response 4-response 1.

Round 2

Reviewer 2 Report

Comments and Suggestions for Authors

The authors have done a good job of changing the specific items mentioned in my review. So, in that sense, the revision was successful.

However, the manuscript still falls short conceptually by not clarifying the implications of these environmental indicator plant studies. Instead, it does a detailed job of saying where those kinds of papers are published and what topics are included.

So, I guess this revised manuscript is now satisfactory but misses an opportunity to say something more compelling. This could be, for example, with more prose in the introduction on what environmental indicator plants tell us, and a bit more in the conclusion on what you have learned from this literature review, with suggestions for redirected research efforts to make up for the revealed gaps and synergies.

Author Response

Coments 1. So, I guess this revised manuscript is now satisfactory but misses an opportunity to say something more compelling. This could be, for example, with more prose in the introduction on what environmental indicator plants tell us,

Response 1: Thank you for pointing this out. We have introduced these new elements in the article’s introduction.

Coments 2.and a bit more in the conclusion on what you have learned from this literature review, with suggestions for redirected research efforts to make up for the revealed gaps and synergies.

Response 2: Thank you for pointing this out. We agree with the comments as indeed, this aspect was missing from our article. Therefore, we have added them now in the Conclusions section.